# Parallel and Efficient Hierarchical k-Median Clustering

**Vincent Cohen-Addad**[*]
Google Research
cohenaddad@google.com

**Silvio Lattanzi**[*]
Google Research
silviol@google.com

**Ashkan Norouzi-Fard**[*]
Google Research
ashkannorouzi@google.com

**Christian Sohler**[†]
University of Cologne
csohler@uni-koeln.de

**Ola Svensson**[*]
EPFL
ola.svensson@epfl.ch

## Abstract

As a fundamental unsupervised learning task, hierarchical clustering has been extensively studied in the past decade. In particular, standard metric formulations as hierarchical $k$-center, $k$-means, and $k$-median received a lot of attention and the problems have been studied extensively in different models of computation. Despite all this interest, not many efficient parallel algorithms are known for these problems. In this paper we introduce a new parallel algorithm for the Euclidean hierarchical $k$-median problem that, when using machines with memory $s$ (for $s \in \Omega(\log^2(n + \Delta + d))$), outputs a hierarchical clustering such that for every fixed value of $k$ the cost of the solution is at most an $O(\min\{d, \log n\} \log \Delta)$ factor larger in expectation than that of an optimal solution. Furthermore, we also get that in for all $k$ simultanuously the cost of the solution is at most an expected $O(\min\{d, \log n\} \log \Delta \log(\Delta dn))$ factor bigger that the corresponding optimal solution. The algorithm requires in $O\left(\log_s(nd \log(n + \Delta))\right)$ rounds. Here $d$ is the dimension of the data set and $\Delta$ is the ratio between the maximum and minimum distance of two points in the input dataset. To the best of our knowledge, this is the first *parallel* algorithm for the hierarchical $k$-median problem with theoretical guarantees. We further complement our theoretical results with an empirical study of our algorithm that shows its effectiveness in practice.

## 1 Introduction

Clustering is a central tool in any large scale machine learning library. The goal of clustering is to group objects into subsets so that similar objects are in the same groups while dissimilar objects are in different groups. There are many different definitions of clustering depending on the description of the objects, the similarity measure between objects, as well as the application. While it is impossible to find the perfect formulation for a clustering problem ([29], see also [13]), the metric version of the problem has received a lot of attention over the last decades and is a central topic in unsupervised learning research.

Here we focus on one fundamental metric clustering problem: the hierarchical Euclidean $k$-median problem. In this formulation, the input objects are described by vectors and the distance (dissimilarity) between objects is measured by their Euclidean distance. More formally, given a set $P \subseteq \mathbb{R}^d$, the objective of the hierarchical $k$-median problem in its classic formulation [33] is to return an ordering

---

[*]Equal contribution

[†]Work was partially done while author was visiting researcher at Google Research, Switzerland.

35th Conference on Neural Information Processing Systems (NeurIPS 2021).

$c_1, \ldots, c_n$ of points and a set of $n$ nested partitions $\mathcal{P}_1, \ldots, \mathcal{P}_n$ of $P$ (i.e., for any $1 \le i \le n$, the $(i-1)$-partition is obtained by merging two clusters in the $i$-partition) such that the following objective is minimized: the maximum over all $1 \le k \le n$ of the ratio of $\sum_{i=1}^{k} \sum_{p \in \mathcal{C}_i} \|p - c_i\|_2$ to the optimum unconstrained $k$-median cost, where $C_k = \{c_1, \ldots, c_k\}$ is the set of centers and $\mathcal{P}_k = \{\mathcal{C}_1, \ldots, \mathcal{C}_k\}$ a $k$-partition. In this paper, we will provide a slightly different guarantee: instead of minimizing the maximum ratio, we show that for every fixed $k$ this ratio is $O(\min\{d, \log n\} \log \Delta)$ in expectation. Our results extend to a slightly weaker bound that holds simultanuously for all $k$. We also remark that the above problem formulation implies a similar bound for the variant of the problem where we do not require an ordering of the centers but instead choose the optimal center for each cluster of the partition.

Hierarchical clustering, thanks to its ability in explaining the nested structures in real world data, has been extensively studied in the computer science literature [27, 26, 22, 43, 9, 14, 41, 44, 5]. The main focus of previous work has been on sequential solutions for the problem and so they are difficult to apply on large data sets. For this reason, several papers recently proposed scalable hierarchical clustering algorithms [27, 26, 22, 43, 5, 39, 34]. Nevertheless most prior work has focused only on scaling the single-linkage algorithm (in fact, efficient MapReduce and Spark algorithms are known for this problem [27, 26, 5, 46]) for which no objective function is clearly specified.

Another closely related area of research focuses on the $k$-median and $k$-means "flat" clustering problems in the distributed setting where several important results are known [3, 6, 18, 35, 1, 10, 23]. The main idea behind these algorithms is to compute in a distributed way some form of small summary of the data (such as coresets). However, all algorithms based on this framework use at least $k + n^\epsilon$ memory per machine (often significantly more) to solve the problem in $O(\frac{1}{\epsilon})$ rounds. Unfortunately, this is too much in our setting where we want to solve the problem also for $k \in \Theta(n)$ (in fact to solve the hierarchical clustering problem we need to compute a solution even for $k = n - 1$). To overcome this problem, in a recent related work [31] Lattanzi et al. provide a parallel algorithm for the hierarchical Euclidean $k$-median problem by dividing the data in two clusters iteratively. Unfortunately, this approach does not provide any theoretical guarantees for the problem (in the experimental section we actually show that it may return low quality solutions) and its round complexity is $O(\log n)$.

**Our Results.** In this work we design the first distributed algorithm for the hierarchical Euclidean $k$-median problem with provable guarantees. In particular, our algorithm returns an approximate solution to the problem using memory $s \in \Omega(\log^2(\Delta + n + d))$, $O\left(\log_s(nd \log(n + \Delta))\right)$ parallel rounds, and $O(nd/s)$ machines, where $\Delta$ is the ratio between the maximum and minimum distance of two points. Note that this implies that our algorithm runs in a constant number of rounds if $s \in n^\delta$ for any constant $\delta > 0$. Interestingly, our algorithm is easily implementable in standard parallel frameworks such as MapReduce [16], Hadoop [45], Spark [47] and Dryad [25] etc. and we analyze it in the standard Massive Parallel Computation (a. k. a. MPC) model [28, 21, 7]. Furthermore, we complement our theoretical analysis with an experimental study in distributed setting where we show that our algorithm is significantly faster than parallel hierarchical [31] and flat solutions [3, 10, 18] for the $k$-median problem.

**Properties, extensions and limitations.** *Work efficiency.* An interesting aspect of our algorithm is that the total running time of our algorithm across all machines is almost linear. In particular, the total running time of our algorithm is $\widetilde{O}(nd)$ so it has almost optimal work efficiency[3]. To the best of our knowledge, our algorithm is also the first approximation algorithm having almost linear running time for the high-dimensional Euclidean $k$-median problem (at the expense of having an expected $O(\min\{d, \log k\} \log \Delta)$-approximation factor).[4]

*Round Lower Bound.* We note here that Theorem 4 in [8] can be easily extended to the Euclidean $k$-median problem[5] and so to the hierarchical Euclidean $k$-median problem. Thus, no distributed

---

[3]In this section, for the sake of simplicity, we use $\widetilde{O}$ notation to hide both $\log n$ and $\log \Delta$ factors. The exact term are provided in the following sections.

[4]We remark that for general metric spaces there is a lower bound of $\Omega(nk)$ total work to achieve a constant approximation guarantee for our problem [37]. Furthermore, we note that finding a constant-factor approximation algorithm for the Euclidean $k$-median problem in almost linear time would imply an asymptotically better approximation algorithm for the closest pair problem which is a long standing open problem in algorithm design. For low-dimensional Euclidean space, near-linear time algorithms are known [11, 12, 30]

[5]The exact same construction and proof work also for the Euclidean $k$-median problem.

algorithm with memory $s$ can output any approximate solution in less than $\log_s n$ rounds. So our memory-round complexity trade-off is asymptotically optimal.

*Simplicity and Efficiency.* Our algorithm is extremely simple to implement. In fact, our results show that it is possible to approximate the hierarchical Euclidean $k$-median problem just by using a sorting algorithm and few aggregation operations. For the same reason, our algorithm is also very parallelizable and efficient.

*Practicality.* Besides our theoretical guarantees our algorithm is also very efficient in practice. We run distributed experiments and we show that the algorithm is more accurate and efficient than previous parallel algorithms. Furthermore, for $k = 10$ our algorithm already outperforms the classic distributed algorithms for flat $k$-median[6] [3, 10, 18] by a factor of 3. For larger $k$ the gap becomes even larger (for example for $k = 1000$ our algorithm is an order of magnitude faster)[7].

*The large $k$ setting.* In a recent paper [8] the authors showed for the first time that it is possible to compute an approximate solution for the Euclidean $k$-means problem using parallel memory significantly smaller than $k$. In particular they provide a bi-criteria algorithm that approximates the solution of the $k$-means problem within a factor $O((\log n \log \log n)^2)$ using $O(k \log k \log n)$ centers, memory-per-machine $s \in \Omega(d \log n)$, and $O(\log_s n)$ parallel rounds. Our algorithm proves that this is also possible for the $k$-median problem. Furthermore our algorithm returns a solution that is not bi-criteria solving an open question in [8].

*Limitations.* Our result has two main limitations. First, it relies on the existence of a good tree embedding that can be computed in a few MPC rounds. As we show in this paper, it is indeed possible to compute a tree embedding of a d-dimensional Euclidean point set with expected distortion $O(d \log \Delta)$ in a constant number of rounds. While there exists a sequential $O(\log n)$-distortion tree embedding construction of point sets of size n of arbitrary metric spaces [20], it remains an interesting open problem as to how this could be implemented in a constant number of rounds. Second, the expected approximation factor of our algorithm is $O(\min\{d, \log n\} \log \Delta)$ and the expectation is for every fixed $k$. Nevertheless, in our experimental analysis, we show that in practice the quality of our solution is comparable to state-of-the-art algorithms while being much faster.

For additional related work we refer the reader to the full version.

## 2 Preliminaries

For two points $p, q$ we use $\|p - q\|$ to denote their Euclidean distance. For a point $p$ and set $C \subseteq \mathbb{R}^d$ we define $\mathrm{DIST}(p, C) = \min_{c \in C} \|p - c\|$, i.e., the distance from $p$ to its closest point in $C$. For two partitions $P_1, P_2$ we say that $P_1$ is nested in $P_2$ if $P_2$ can be obtained from $P_1$ by merging two or more parts of $P_1$. Given a point set $P \subseteq \mathbb{R}^d$, a set of centers $C = \{c_1, \ldots, c_k\}$, and a partition $\mathcal{P}$ of $P$ into $k$ parts $\{\mathcal{P}_1, \ldots, \mathcal{P}_k\}$, its $k$-median cost is $\mathrm{COST}(P, C, \mathcal{P}) = \sum_{i=1}^{k} \sum_{p \in \mathcal{P}_i} \mathrm{DIST}(p, c_i)$. The points in $C$ are called the *cluster centers*. We let $\mathrm{OPT}(P, k) = \min_{C \subseteq \mathbb{R}^d, |C|=k} \min_{\mathcal{P}, |\mathcal{P}|=k} \mathrm{COST}(P, C, \mathcal{P})$ denote the cost of an optimal $k$-median solution.

In this paper, we focus on a slight variation of the classic Hierarchical Euclidean $k$-median problem[8], which for a given point set $P \subseteq \mathbb{R}^d$ asks to find an (ordered) sequence of centers $C = \{c_1, \ldots, c_n\}$ together with a collection of $n$ nested partitions $\Pi = \{\mathcal{P}_1, \ldots, \mathcal{P}_n\}$ of $P$, such that $\mathcal{P}_i$ partition the entire space and contains $i$ parts. We say that an algorithm for the Hierarchical Euclidean $k$-median problem has an approximation guarantee of $\alpha$ if, for every $k = 1, \ldots, n$, the output $C, \Pi$ defines an $\alpha$-approximate solution for $k$-median. That is, for every $k \in \{1, 2, \ldots, n\}$ $\mathrm{COST}(P, \{c_1, \ldots, c_k\}, \mathcal{P}_k) \leq \alpha \cdot \mathrm{OPT}(P, k)$. Similarly, if the algorithm is randomized then we say that it has an approximation guarantee of $\alpha$ if the guarantee holds in expectation over the random hierarchical clustering $C, \Pi$ output by the algorithm: for every $k \in \{1, 2, \ldots, n\}$, $\mathbf{E}[\mathrm{COST}(P, \{c_1, \ldots, c_k\}, \mathcal{P}_k)] \leq \alpha \cdot \mathrm{OPT}(P, k)$.

---

[6]Note that flat clustering is a sub-problem of hierarchical Euclidean $k$-median problem, in fact our algorithm solve the problem for all $k$ at the same time

[7]Note that the large $k$ has many practical applications in spam and abuse [38, 42], near-duplicate detection [24], compression or reconciliation tasks [40]

[8]We note here that our results naturally extend to the Prefix Euclidean $k$-median problem [36], where clusters are not required to be nested.

For simplicity of presentation, here we assume that $P \subseteq \{0, \ldots, \Delta\}^d$ and then we explain how to remove this assumption in our theoretical analysis in Appendix. The $k$-median problem can also be formulated for discrete metric spaces $M = (P, \text{DIST}_M)$, where $P$ is a finite set of points and $\text{DIST}_M$ is a metric. For a subset $C \subseteq P$ of the points, we also let $\text{DIST}_M(p, C) = \min_{c \in C} \text{DIST}_M(p, c)$ be the distance from the point $p$ to its closest point in $C$. The objective of the metric $k$-median problem is then to find a subset $C \subseteq P$ of size $k$ such that $\text{COST}_M(P, C) = \sum_{p \in P} \text{DIST}_M(p, C)$ is minimized. The hierarchical $k$-median problem is analogously defined.

A (discrete) metric space $(P, \text{DIST}_T)$ is called a *tree metric*, if there exists a positively weighted tree $T = (P, E, w)$ such that for all pairs $p, q \in P$ we have that $\text{DIST}_T(p, q)$ equals the shortest path distance between $p$ and $q$ in $T$. We will be mostly interested in tree metrics that are defined by *restricted hierarchically well separated trees.* [9]

**Definition 2.1** *A* restricted $\ell$-hierarchically well separated tree (RHST) *is a positively weighted rooted tree such that all the leafs are at the same level, all edges at the same level have the same weight, and the length of the edges decreases by a factor of $\ell$ on every root to leaf path.*

Throughout the paper, we consider RHSTs with $\ell = 2$, i.e., 2-RHSTs. We therefore simplify notation and sometimes write RHST for 2-RHST. In the next section, we describe how to embed our point set $P$ in Euclidean space into the leaf of a RHST. Formally, a metric embedding between two metric spaces $(P, \text{DIST})$, $(P', \text{DIST}')$ is an injective mapping $f : P \to P'$. To simplify notation and to be consistent with this definition (and the definition of a metric) we consider instances that contain no two points within distance 0. However, all our arguments and algorithms generalize to instances that may have several identical points.

**MPC model.**   We design algorithms for the MPC model [28, 21, 7] that is considered de-facto the standard theoretical model for large-scale parallel computing. Computation in MPC proceeds in synchronous parallel *rounds* over multiple machines. Each machine has memory $s$. At the beginning of a computation, data is partitioned across the machines. During each round, machines process data locally. At the end of a round, machines exchange messages with a restriction that each machine is allowed to send messages and also receive messages of total size $s$. The efficiency of an algorithm in this model is measured by the number of rounds it takes for the algorithm to terminate and by the size of the memory of every machine. In this paper we focus on the most challenging and practical regime of small memory. In particular, we only assume to have $s \in \Omega(\log^2(n + \Delta + d))$.

## 3   A Work Efficient Sequential Algorithm

In this section we introduce a work efficient sequential algorithm and then in the next section, we show how to parallelize it in the MPC model. From a high level perspective our algorithm is based on embedding the input point set into the leaf of a RHST and then solving the problem optimally for the embedded points. In order to obtain the embedding, we choose a (standard) random quad-tree embedding of the input points into a hierarchically well separated tree. We then show that a simple greedy algorithm solves the problem optimally on the induced tree metric. Unfortunately, a naive implementation of our greedy algorithm would result in large parallel and sequential running time. So we modify our greedy algorithm and we show that it can be implemented using only few aggregation operations combined with sorting.

### 3.1   Quadtree Embedding into a 2-RHST

The first step of our algorithm is to embed the points in a restricted 2-hierarchically separated tree. Interestingly, we observe that using standard embedding techniques it is possible to embed all the points in the datasets in a 2-RHST by incurring only a small distortion. Furthermore, in this construction every point can compute its own position in the embedding independently just by knowing its own coordinated and a random shift $r$ that is applied to all the points.

Given that the construction of the embedding is standard we defer it to the full version. Now we present formally the main property of the embedding that will be useful for our algorithm. Before

---

[9]Our definition is slightly more restricted than the standard notion of $\ell$-HST (see, for example, [4]), because we require that all edges at the same level have the same cost (rather than all descendants of a node) and that the cost decreases by the same factor.

stating the property we need to introduce some additional notation. For two points $p, q$ in the RHST $T$, we use $\text{DIST}_T(p, q)$ to denote their (shortest-path) distance. Similarly we let $\text{DIST}_T(p, C) = \min_{c \in C} \text{DIST}_T(p, c)$. We now define $\text{COST}_T(P, C) := \sum_{p \in P} \text{DIST}_T(p, C)$ to be the cost of a set of centers $C$ with respect to the tree metric. We use $\text{OPT}_T(P, k) := \min_{C \subseteq \mathbb{R}^d, |C|=k} \text{COST}_T(P, C)$ to denote the cost of optimal solution with respect to a given RHST $T$. Recall that $\text{OPT}(P, k)$ denotes the cost of an optimal solution with respect to the original Euclidean distances. Now we are ready to state the main result of this section whose proof is presented in the full version.

**Theorem 3.1** *Let $P \subseteq \{0, \dots, \Delta\}^d$ be a point set. There exists a procedure that constructs a 2-RHST tree $T$ in time $O(nd \log \Delta)$ such that for the its optimum solution $C^*_{T,k}$ using $k$ centers, we have $\mathbf{E}[\text{COST}(P, C^*_{T,k})] = O(d \cdot \log \Delta) \cdot \text{OPT}(P, k)$. Furthermore all the input points are mapped to leaves of the RHST. In addition, we have that $\mathbf{E}[\max_k \frac{\text{COST}(P, C^*_{T,k})}{\text{OPT}(P,k)}] = O(d \cdot \log \Delta \log(dn\Delta))$.*

## 3.2 Optimal Algorithms for $k$-Median on $2$-RHST

In this subsection, we design two optimal sequential algorithms for hierarchical $k$-median on a 2-RHST metric. This combined with our 2-RHST embedding will give us an efficient approximation algorithm. We first show that a simple greedy algorithm (see Algorithm 1) finds the optimum solution. The greedy algorithm chooses, at each step, the point that leads to the largest decrease in the cost. Then we modify the algorithm to be more amenable to parallel implementation and to run in almost linear time.

### 3.2.1 A Greedy Algorithm for $2$-RHST

We start by providing the pseudo-code for our algorithm in Algorithm 1 and state the main property of our greedy algorithm (whose proof is deferred to the full version ).

**Theorem 3.2** *For any set of points $P \subseteq \{0, \dots, \Delta\}^d$ and distance function $\text{DIST}_T$ defined by an 2-RHST $T$, Algorithm 1 returns an optimum solution for the hierarchical $k$-median problem.*

*Proof.*(Sketch) First observe that by the cluster assignment in line (8) the returned partitions $\{\mathcal{P}_1, \dots, \mathcal{P}_n\}$ are nested. It remains to show that for every $k$, the partition $\mathcal{P}_k$ with centers $c_1, \dots, c_k$ is optimal. The proof is by induction on the height of the tree $T$. For the base case, when $T$ is of height 0, the statement is clear: In that case the instance consists of a single point so $\text{COST}_T(P, S) = 0$ if $S \neq \emptyset$.

For the inductive step we make two fundamental observations. First, we observe that in a 2-RHST metric the cost of a solution can be decomposed as a sum of the cost of individual subtrees. Second, we also show that the cost of each subtree is roughly independent of the choice made by the algorithm in other subtrees. So combining these two facts, with the fact that greedy is optimal for all the subtrees of we can conclude our inductive arguments. $\square$

---

**Algorithm 1** GREEDY alg. for hier. $k$-median on 2-RHST

---

**Input:** Set of points $P$, cost function $\text{COST}_T$ defined
    by a 2-RHST $T$
1: Set $S_0 \leftarrow \emptyset$
2: Set $P_0 \leftarrow \{P\}$
3: Label all internal nodes of the RHST as unlabelled
4: **for** $i = 1$ to $n$ **do**
5:    Let $c_i = \text{argmin}_{x \in P} \text{COST}_T(P, x \cup S_{i-1})$
6:    Label the highest unlabelled ancestor of $c_i$ with $c_i$
7:    $S_i$ is obtained by adding $c_i$ to $S_{i-1}$
8:    Define $\mathcal{P}_i$ as the clustering obtained by assigning all points to the cluster centered at their closest labelled ancestor.
9: **end for**
**Output:** return $c_1, \dots, c_n, \mathcal{P}_1, \dots, \mathcal{P}_n$

---

### 3.2.2 Quasi Linear Time Algorithm

In this subsection we show how to change the previously presented algorithm (Algorithm 1) to obtain an almost linear-time algorithm that can be easily parallelized. The algorithm will output a list of centers and an implicit description of a list of nested partitions. The list of centers as well as the partitions will be identical to that of the previous algorithm. Hence, we will focus on the optimality of the centers wrt. the hierarchical $k$-median problem.

As a first step we modify Algorithm 1 to reduce its sequential dependencies. The downside of Algorithm 1 is that the selection to open the $i$-th point depends on the previously chosen points, i.e., $S_{i-1}$. The main idea is to remove this dependency and to make the decision dependent on other parameters that can be efficiently precomputed. Before describing our result we introduce some additional notation. We say that a node in the (2-RHST) tree is at level $i$ if the subtree rooted at this node is of height $i$. We remark that with this convention a leaf is at level $0$ and the root is at the maximum level $h$. Let $S$ be a set of centers and $x$ be a point in $P \setminus S$. We refer to the ancestor of $x$ at level $i$ by $a_i(x)$ and, with some abuse of notation, we denote with $p_i(x)$ the number of points in the subtree of $a_i(x)$ for $0 \le i \le h$. We also write $A_i(x) \subseteq P$ for the set of points in the subtree of node $a_i(x)$. Finally, we let $\ell(x) \in \{1, \ldots, h\}$ be the highest level $\ell$ for which $S \cap A_\ell(x) = \emptyset$. For clarity, when it is clear from the context, we drop the $x$ and use $a_i, A_i, p_i, \ell$.

Our first key observation is that the distance of a point $x$ to its nearest center only depends on $\ell$. This is true because i) the distance of $x$ to the point $y \in S \cap A_{\ell+1}$ is exactly twice the distance of $x$ to $a_{\ell+1}$, which is $2 \sum_{i=0}^{\ell-1} 2^i = 2(2^\ell - 1) = 2^{\ell+1} - 2$. ii) The distance to any other point $w \in S$ is at least $2^{\ell+2} - 2$ since their common ancestor is at least at level $\ell + 1$. Therefore,

**Observation 3.3** *The distance* $\mathrm{DIST}(x, S)$ *of a point $x$ to its nearest center in $S$ only depends on $\ell(x)$. More precisely,* $\mathrm{DIST}_T(x, S) = 2^{\ell(x)+1} - 2$.

In addition, we observe that the benefit of opening $x$ (denoted by $\mathrm{BENF}(x)$) only depends on the number of nodes in close subtrees, $p_1, \ldots, p_\ell$. In fact opening $x$ only affects the cost of the points in $A_\ell$. This is true because any point $w$ that is in $A_{\ell+1} \cap S$ is at least as close to $x$ as to any other point in $A_{\ell+1}$ and not in $A_\ell$. Now, consider a point $y$ in $A_i$ for $0 \le i \le \ell$. The cost of $y$ after opening $x$ is $2^{i+1} - 2$, and before opening $x$ its cost was $2^{\ell+1} - 2$. Furthermore, we can precisely compute the number of points in $A_i$ and not in $A_{i-1}$, it equals $p_i - p_{i-1}$. So we get:

**Lemma 3.4** *The benefit of opening point $x$ only depends on $p_1(x), \ldots, p_{\ell(x)}(x)$ and is* $\mathrm{BENF}(x) = \sum_{i=0}^{\ell(x)} (p_i(x) - p_{i-1}(x)) \cdot (2^{\ell(x)+1} - 2^{i+1})$, *where for simplicity we assume $p_{-1}(x) = 0$.*

These simple observations significantly reduces the dependency of computing the benefit of opening a point $x$ from the set of centers $S$, which is the main step of Algorithm 1. In fact, the $\mathrm{BENF}(x)$ value does not depend on the full structure of $S$ but only on $\ell(x)$. Therefore, we can compute $\mathrm{BENF}(x)$ for all $\ell$ before opening the centers and use only those values to make our selection of the centers. We denote this by $\mathrm{BENF}(x, \ell)$, i.e., $\mathrm{BENF}(x, \ell) = \sum_{i=0}^{\ell} (p_i(x) - p_{i-1}(x)) \cdot (2^{\ell+1} - 2^{i+1})$, for all points $x$ and $0 \le \ell \le h$.

Now, to select the centers we can design an alternative algorithm based only on the $\mathrm{BENF}(x, \ell)$ values. In fact we can sort all the $\mathrm{BENF}(x, \ell)$ in an ordered list $L$ and go over them from the highest value to the lowest. We show that it is sufficient to prune the list $L$ by removing a pair $(x, \ell)$ in the following two cases:
– There is another point with larger benefit in $A_\ell$. In this case we need to update our sorted list by removing $(x, \ell)$ from the ordering $L$. In general to deal with this case for each subtree rooted at level $\ell$ for all possible $\ell$ we preserve only one pair $(x, \ell)$ with highest benefit.
– For point $x$, collect all pairs $(x, \ell)$ that remained after the previous pruning step. Identify the pair with the maximum value of $\ell$ and prune all other collected pairs.

After these two steps, the ordering $L$ can be used to obtain our final solution for Hierarchical $k$-median (resp. $k$-median) by returning the centers in the same order as in $L$ (resp. the top $k$ centers of $L$). The pseudo-code is presented in Algorithm 2.

Importantly, we show that the output sequence of centers and the implicitly defined partitions of Algorithm 2 are the same as the output of Algorithm 1. This is intuitively true because the pruning rules identify at which level $x$ is a good candidate for the greedy algorithm and then we sort the $x$ based on their score at that level. Note that in this algorithm we only output the partition implicitly because outputting them explicitly will take $\Omega(n^2)$ time. The full proof of this argument is presented in the full version.

**Theorem 3.5** *Algorithm 2 finds an optimum solution for the hierarchical $k$-median problem on RHSTs in time $O(n \log^2(\Delta + n))$.*

# 4 Euclidean $k$-Median in the MPC Model

In this section we present our main results in the
MPC model. The algorithm that we use is the
same as Algorithm 2, but we implement each
step in the MPC model. Our algorithm is based
on basic operations: summation, computing the
maximum and sorting a list of elements. All
these operations can be done efficiently in the
MPC model using [21]. We provide the details
on the implementation of our algorithm along
with its theoretical performance guarantees.

**Theorem 4.1** *There is a distributed algorithm
that uses memory $s \in \Omega(d \log(\Delta) + \log^2(\Delta + n))$ and computes a solution for the hierarchical
Euclidean k-median problem such that for every fixed k the k-clustering of the hierarchy has
an expected approximation factor of $O(d \log \Delta)$.
Furthermore, the expected maximum approximation factor (over all k) is $O(d \log \Delta \log(\Delta dn))$.
The algorithm uses $O(\log_s(n \log \Delta))$ parallel
rounds and $O((n \log \Delta)/s)$ machines in the
MPC model. Furthermore the total running
time of the algorithm across all the machine
is $O(nd \log(\Delta + n))$.*

*Proof.* Our distributed algorithm consists of a
parallel implementation of Algorithm 2 that we
run on the embedding described in Section 3.1.
This gives us an expected approximation factor
of $O(d \log \Delta)$, because from Theorem 3.5 we
know that Algorithm 2 solves the problem optimally on 2-RHST and from Theorem 3.1 we
know that we loose a factor $O(d \log \Delta)$ in expectation in our embedding step.

---

**Algorithm 2** Quasi-linear alg. for hier. $k$-median on RHST

---

**Input:** Set of points $P$, cost function $\text{COST}_T$ defined by a 2-RHST $T$
1: $\forall x \in P, 0 \le \ell \le h$, compute $\text{BENF}(x, \ell)$
2: **for** $0 \le \ell \le h$ **do**
3:      **for** subtree $T' \subseteq T$ rooted at level $\ell$ **do**
4:          Let $x \in T'$ be the point maximizing $\text{BENF}(x, \ell)$
5:          **for** $y \in T'$ **do**
6:              $C_{(x,\ell)} \leftarrow C_{(x,\ell)} \cup \{y\}$
7:              If $y \neq x$ delete $\text{BENF}(y, \ell)$
8:          **end for**
9:      **end for**
10: **end for**
11: For every $x$ keep only the pair $(x, \ell)$ with maximum $\ell$ and its cluster $C_{(x,\ell)}$.
12: Sort all pairs $(x, \ell)$ according to $\text{BENF}(x, \ell)$ and let $(c_i, \ell_i)$ be the pair in position $i$.
13: Implicitly define the $i$-th partition $\mathcal{P}_i = \{P_1, \ldots, P_i\}$ as follows: $P_j = C_{(c_j, \ell_j)} \setminus \cup_{z=j+1}^i C_{(c_z, \ell_z)}$ for $1 \le j \le i$.
**Output:** Return the ordering of the points $c_1, \ldots, c_n$ as centers and the clusters $C_{(c_i, \ell_i)}$ $\forall i$ to implicitly define the partitions.

---

The details for the distributed implementation of this algorithm in the MPC model is as follows. In
the implementation we need to be able to compute the embedding, compute $p_\ell(x)$ for all points $x$,
compute $\text{BENF}$, discard the useless $\text{BENF}$ pairs and sort the values. First, note that we can compute the
embedding in parallel for each of the points because the embedding only depends on their coordinates
and a uniform random shift. Then to compute $p_\ell(x)$ and $\text{BENF}$ for all nodes we just need to be able to
compute weighted sums efficiently. This can be done using memory $s \in \Omega(d \log \Delta + \log^2(\Delta + n))$ in
$O(\log_s(n \log \Delta))$ MPC rounds with $O((n \log \Delta)/s)$ machine and total running time of $O(nd \log \Delta)$
using the algorithm in [21]. Then we need to sort the computed $\text{BENF}$ values which can be done
similar to be previous step using [21] with the same bounds as before. To apply the filter we just
need to be able to find the maximum in a list of values and we can again do this using memory
$s \in \Omega(\log^2(\Delta + n))$ in $O(\log_s(n \log \Delta))$ MPC rounds using the algorithm in [21]. Note that after
computing the maximum we can also construct the sets $C_{(x,\ell)}$ by outputting as $C_{(x,\ell)}$ all nodes
sharing the same ancestor of $x$ at level $\ell$, and this can be done in $O(1)$ rounds as well. Finally we can
also do sorting with the same bound using the same techniques with total running time of $O(n \log n)$.
□

We remark that we can speed up the algorithm for datasets of large dimension, by reducing the
dimension from $d$ to $O(\log n)$ by losing a small constant factor in the approximation ratio[15] and
achieve the following result, the details are presented in the full version. We also remark that the
guarantee on the expected maximum approximation ratio (over all $k$) from Theorem 3.1 carries over
to this theorem and the following corollary.

**Corollary 4.2** *There is a distributed algorithm that uses memory $s \in \Omega(\log^2(\Delta + n + d))$ and computes a solution for the hierarchical Euclidean k-median problem such that*

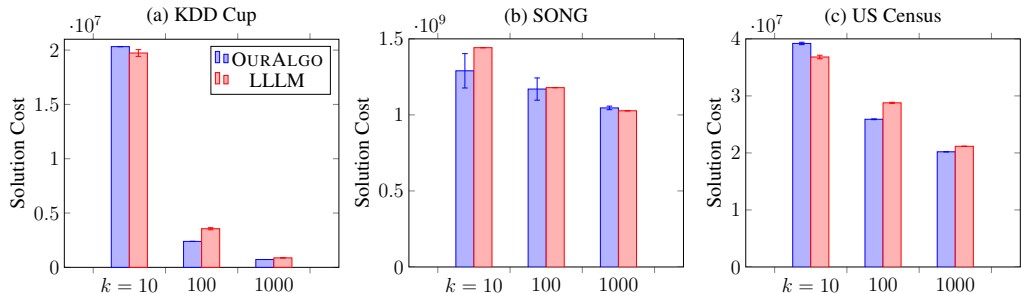

Figure 1: In this plot, we compare our algorithm (OURALGO), and LLLM. The results are presented in for KDD Cup (a), SONG (b), US Census (c). All the numbers are reported for 5 runs (average value and variance).

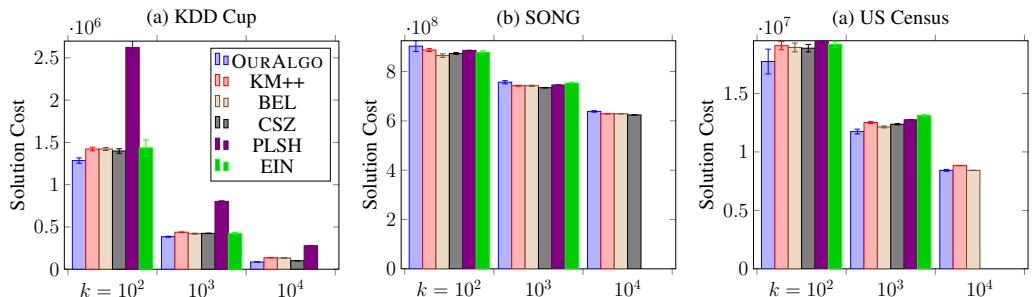

Figure 2: In this plot, we compare our algorithm (OURALGO), $k$-median++ seeding (KM++ Seed), BEL, PLSH, and EIM. The results are presented in for KDD Cup (a), SONG (b), Us Census (c). All the numbers are reported for 5 runs (average value and variance).

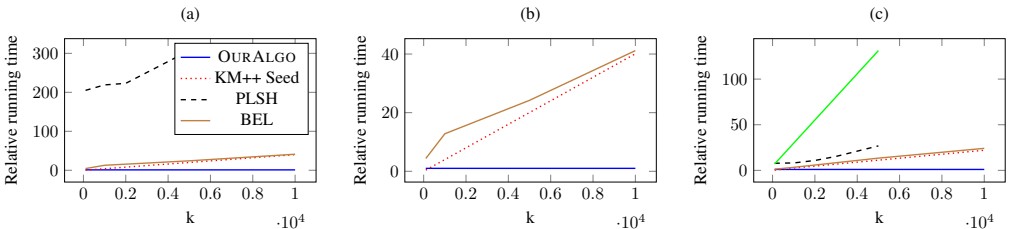

Figure 3: Figure (a), (c) compares the running time for our algorithm (OURALGO), $k$-median++ seeding (KM++ Seed), BEL, PLSH, and CSZ on HIGGS and US Census datasets, respectively. In (b) we focus on our algorithm and $k$-median++ seeding on HIGGS dataset to further highlight their differences.

*for every fixed $k$ the $k$-clustering of the hierarchy has an expected approximation factor of $O(\min(d, \log n) \log \Delta \log(nd\Delta))$. The algorithm uses $O(\log_s(nd \log(n + \Delta)))$ parallel rounds and $O(nd \log(n + \Delta)/s)$ machines in the MPC model. Furthermore the total running time of the algorithm across all the machine is $O(nd(\log(n + \Delta)))$. Finally the algorithm returns a solution also for the prefix $k$-median clustering and the hierarchical $k$-median clustering problems.*

## 5 Empirical Evaluation

In this section we empirically evaluate our algorithm with both hierarchical and non-hierarchical algorithms. We report the expected value and the variance over 5 runs for all the randomized algorithms. We evaluate the algorithms on the following data sets from UCI Machine Learning Repository [17]; KDD Cup (a.k.a Intrusion, $n = 31,029; d = 35$),YearPredictionMSD (a.k.a SONG, $n = 515,345; d = 90$),US Census Data (1990) ($n = 2,458,285; d = 29$),and HIGGS ($n = 11,000,000; d = 28$).

## 5.1 Distributed Hierarchical Algorithms

We compare our algorithm with LLLM [32] - an algorithm for hierarchical $k$-median problem in distributed setting. In Fig. 1 we compare the quality of the produced solution. We observe that the algorithms are comparable and our algorithm outperforms LLLM by $2 - 3\%$ on average. The running time of the the LLLM algorithm divided by the running time of our algorithm is presented in Table 1. Our algorithm is significantly faster (up to a factor

| Dataset | m=1x | 10x | 100x |
|---------|------|------|------|
| KDD Cup | 9.58 | 13.5 | 26.9 |
| SONG | 2.32 | 5.26 | 19.3 |
| US Census | 1.72 | 3.10 | 6.71 |

Table 1: Distributed hierarchical algorithms running time comparison for various number of machines. The running time of the LLLM baseline divided by the running time of our algorithm with the same amount of machine used.

$10 - 26$ depending on the dataset) and the difference increases with number of the machines used.

## 5.2 Distributed Algorithms

Our algorithm can be also used for the (non-hierarchical) $k$-median problem. In order to assess its quality, we compare it with sequential and distributed algorithms for the (non-hierarchical) $k$-median problem. i) We compare the quality of the computed set of centers and the sequential running time with both sequential and distributed algorithms for the k-median problem, ii) we compare the running time with distributed algorithms in a distributed setting, iii) we run our algorithm on massive datasets of with tens of billions of nodes and report the speedup gained by using more machines. Notice that the baselines do not scale to this size of data, therefore we cannot compare the performance.

**Baselines.** We compare our algorithm (without dimensionality reduction) with a well-known sequential algorithm: $k$-median++ seeding; and four distributed algorithms: PLSH [8], EIM [19], BEL [2], CSZ [10]. Description of these algorithms and the details of the setting used for running the algorithms are provided in the full version.

**Quality and Sequential Running Time Comparison.** Our algorithm is $40, 41, 300$ times faster than $k$-median++, BEL, and PLSH, respectively for census dataset for $k = 10,000$. Also it is significantly faster than CSZ [10]. In figure Fig. 3 we compare the running time of these four algorithms for different $k$[11] (recall that we compute the solution for all $1 \le k \le n$ at the same time). The plots are similar for other datasets and are presented in the full version. Let us focus on the quality of the solution. The results are presented in Fig. 2. Note that due to the slow running time and memory consumption of the PLSH, CSZ, and EIM we were not able to run it in some cases. The results suggest that our algorithm provides comparable results.

**Distributed Running Time Comparison.** Now we focus on the running time of OURALGO, BEL, EIM, and CSZ algorithms in distributed setting for HIGGS dataset. We compare the running time of the baselines with our algorithm with the same amount of machine used for k=10 to $10,000$. We present the running time of the baselines divided by the running time of our algorithm. The results are presented in the full version. Our algorithm is significantly faster than all baselines. For small values of $k$, e.g: $k = 10$, we are faster by a factor $3.2 - 60.2, 50 - 182$, and $65 - 1380$ compared to BEL, EIM, and CSZ, respectively. Moreover, for $k = 1,000$ we are $12.08 - 71.2, 2082 - 3091$, $46 - 941$ times faster compared to BEL, EIM, and CSZ, respectively. Also for $k = 10,000$ we are $41.17 - 126$ and $171 - 708$ times faster than BEL and CSZ, respectively.

**Scalability on Large Datasets.** To provide an empirical scalability analysis of our algorithms over large datasets, we expanded the HIGGS dataset so as to experiment on datasets of size almost 100 millions, 1 billion, and 10 billions. More details can be found in the full version of the paper.

---

[10]The missing values are due to the slow running time of CSZ.

[11]EIM is slower that rest of the algorithms and its running time is presented in the Appendix.

# 6    Conclusions

We present the first distributed approximation algorithm for Euclidean Hierarchical $k$-Median problem in the MPC model. Interestingly our algorithm even works in the setting where each machine has very limited memory $s \in \Omega(\log^2(\Delta + n + d))$ and it is work efficient. In the future, it would be interesting to obtain similar results for other clustering problems and to improve the approximation factor of our algorithm. We believe that the core ideas used in this work can extend to the general $k$-clustering problem (e.g., $k$-means and higher powers). To be more precise, if we are given a RHST that increases the distances by at most a factor of $g(n)$, then we believe that our approach construct a $g(n)$ approximate solution, for any function $g$. The difficulty here is to construct such RHST.

## Acknowledgments and Disclosure of Funding

This research was partially supported by the Swiss National Science Foundation projects 200021-184656 "Randomness in Problem Instances and Randomized Algorithms".

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
