(see Appendix E for a more detailed description of the generation process). As depicted in Fig. 5 in Appendix K, the speed-ups achieved by our algorithm is significant when increasing the number of machines.

---

[10]The missing values are due to the slow running time of CSZ.

[11]EIM is slower that rest of the algorithms and its running time is presented in the Appendix.

# 6   Conclusions

We present the first distributed approximation algorithm for Euclidean Hierarchical $k$-Median problem in the MPC model. Interestingly our algorithm even works in the setting where each machine has very limited memory $s \in \Omega(\log^2(\Delta + n + d))$ and it is work efficient. In the future, it would be interesting to obtain similar results for other clustering problems and to improve the approximation factor of our algorithm. We believe that the core ideas used in this work can extend to the general $k$-clustering problem (e.g., $k$-means and higher powers). To be more precise, if we are given a RHST that increases the distances by at most a factor of $g(n)$, then we believe that our approach construct a $g(n)$ approximate solution, for any function $g$. The difficulty here is to construct such RHST.

## Acknowledgments and Disclosure of Funding

This research was partially supported by the Swiss National Science Foundation projects 200021-184656 "Randomness in Problem Instances and Randomized Algorithms".

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

# A  Additional Related Works

**Additional Related Work.**  The $k$-median problem has also been extensively studied for tree metrics (see e.g. [54, 11] and references therein) where exact polynomial-time algorithms exist. We remark that we focus on the special case of hierarchically separating trees, which we can exploit to get faster algorithms. Indeed previous work on tree metrics only give nearly linear running time if $k$ is a constant.

Somewhat related to our work is the quadtree-based coreset construction in [27]. The resulting coreset can be computed in $O(n \log \Delta)$ time and may be interpreted as a bi-criteria approximation, i.e., it uses more than $k$ centers to achieve an approximation. We also believe that the construction in [13], which is based on randomly shifted quadtrees, can be modified to efficiently get a bi-criteria approximation. A closely related problem is the $k$-medoid problem, that is as $k$-median with the constraint that the centers are restricted to be from the input points. For this problem, a number of algorithms have been developed (see, for example, [37, 47]), but unfortunately they are sequential and their running times do not scale in our setting.

We also remark that the question of obtaining an efficient distributed algorithm for hierarchical clustering has been studied in a series of recent paper [8, 40, 57]. Although no result was known for hierarchical Euclidean median before this work. Finally, we note that the HSTs have been exploited to design efficient algorithm for fair clustering [2], it is an interesting open question to extend our results in their settings.

Interesting results are also known for the robust variant of $k$-median with outliers [16] and for the $k$-means problem [3].

# B  Description of Baseline Algorithms and Experiments Setup

In this section for the sake of completeness we present a general overview of the baseline algorithms. Notice that this provides the reader with some of the ideas used in these algorithms. The implementation used is exactly the algorithm presented in the respective papers. Notice that we used the $k$-median++ seeding as the offline algorithm used for both EIM and BEL algorithms since its performs well in practice and its running time is of $O(nkd)$. Moreover, it suffices for the machines to have memory $1G$ for all the distributed algorithm for HIGGS dataset.

- Our implementation of $k$-median++ seeding - an adaptation of $k$-means++ seeding [1] for the $k$-median problem - works as follow. The algorithm first samples a point uniformly at random from the input points (i.e. $P$). Then in the next $k - 1$ iterations, it picks the next center from $P$ with probability proportional to the distance to the nearest of the current centers. It is known that this seeding produces an $O(\log k)$-approximate solution. In the experiments we use our own implementation of $k$-median++.

- PLSH [12] algorithm (code by authors of PLSH). This algorithm first finds a bicriteria solution based on LSH and then samples a set of $k$ elements from it. In the implementation to obtain $k$ centers from the bi-criteria approximation, the centers are chosen uniformly at random from it. We thank the authors for helping us with experiments using their code.

- Our implementation of EIM [25] algorithm. This algorithm has two phases. The first phase starts with two sets $S = \emptyset, H = \emptyset, R$, where $R$ at the beginning is equal to the set of input points. In each iteration of the first phase two set of input elements are sampled and added to $H, S$. Then for each point in $S$ the closets point in $H$ is computed and based on that a threshold value is computed. Then for each point currently in $R$ the closest point to any point in $H$ is computed and the point is removed it the computed minimum distance is lower than the threshold. The iterations are repeated till size of $R$ is below a certain value depending on $k, n$. Then a weighted coreset over $H \cup R$ is computed and the final solution is computed based on it using any offline algorithm.

- Our implementation of BEL [4] algorithm. This algorithm divides the input into $m$ part, where $m$ is the number of machines. Then runs any offline algorithm and creates an weighted coreset based on the offline algorithms solution.

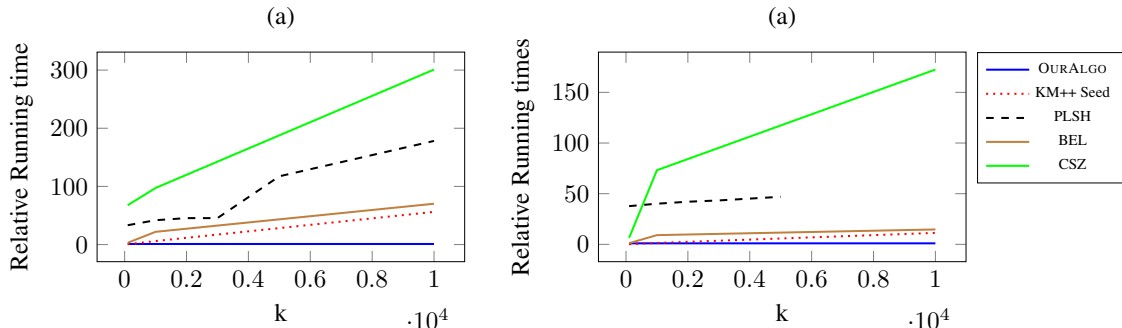

Figure 4: comparison between the running time of our algorithm with $k$-median++ seeding, BEL, CSZ, and PLSH for SONG (a) and KDDCUP (b) datasets.

comparison with BEL

| k \ m | 1x | 10x | 100x |
|---|---|---|---|
| 10 | 3.2 | 12.45 | 60.2 |
| 25 | 3.75 | 12.61 | 61.3 |
| 50 | 4.01 | 14.7 | 62.0 |
| 100 | 4.36 | 15.3 | 62.4 |
| 1000 | 12.08 | 25.1 | 71.2 |
| 10000 | 41.17 | 87.6 | 126 |

comparison with EIM

| k \ m | 1x | 10x | 100x |
|---|---|---|---|
| 10 | 50.4 | 60.5 | 182 |
| 25 | 90.6 | 72.2 | 192 |
| 50 | 144.4 | 110 | 262 |
| 100 | 258 | 330 | 452 |
| 1000 | - | 2082 | 3091 |
| 10000 | - | - | - |

comparison with CSZ

| k \ m | 1x | 10x | 100x |
|---|---|---|---|
| 10 | 65.3 | 276.6 | 1380 |
| 25 | 59.33 | 278 | 1483 |
| 50 | 69.3 | 258 | 1375 |
| 100 | 41.67 | 176.67 | 1341 |
| 1000 | 46.67 | 145 | 941.67 |
| 10000 | 171.5 | 233 | 708.33 |

Table 2: Distributed running time comparison for various number of centers in the solution $k$ and number of machines $m$. The running time of the BEL (first table), EIM (second table), and CSZ (third table) baselines divided by the running time of our algorithm with the same amount of machine used. The empty fields are due to slow running time of the baseline.

- Our implementation of LLLM [41] algorithm. This algorithm is distributed and hierarchical, it divides each cluster into two until the size of all the clusters are one. It guarantees that the number of parallel rounds is $O(\log n\Delta)$ but it does not provide a theoratical guarantee.

## C  Further Sequential Comparisons

Fig. 4 presents comparisons between the running time of our algorithm with $k$-median++ seeding. BEL, and PLSH for KDD Cup and SONG datasets. The running time of our algorithm is at least a factor 125, 437, and 3121 faster than EIM even for $k = 100$ for KDDCUP, SONG, and Census datasets, respectively. The gap increases by increasing $k$, therefore we do not add EIM results to the plots. These results are expected and is supported by the theoretical analysis of our algorithm.

## D  Coreset Comparisons

One of the ideas to improve the quality of the solution discussed in various works (e.g., [4]) is to construct a coreset, and then apply possibly slower algorithms that provide solutions with better

| dataset \ k | 100 | 1000 | | k \ dataset | 100 | 1000 |
|---|---|---|---|---|---|---|
| KDD CUP | 1136790 | 371302 | | KDD CUP | 1311390 | 436409 |
| Song | 762199000 | 648547000 | | Song | 805580000 | 668776000 |
| Census | 15123100 | 10104100 | | Census | 16761500 | 10679170 |

Table 3: In this table, we compare the quality of the solution of our algorithm (left table) and BEL(right table) after applying the greedy improvements. All the numbers are reported for 5 runs.

| dataset \ k | 100 | 1000 |
|---|---|---|
| KDD CUP | 3.52 | 3.91 |
| Song | 1.11 | 4.53 |
| Census | 1.99 | 12.6 |

Table 4: In this table, we compare the running time if BEL divided by the running time of our algorithm after applying the greedy improvements. All the numbers are reported for 5 runs.

quality on the coreset. One such algorithm is k-median++ followed by greedy improvement steps likes Lloyds algorithm. In the BEL algorithm [4], authors after constructing the coreset use this technique to improve the quality of the solution. Notice that since the size of the coreset is smaller than the original input, the running time of these greedy steps are better than applying the same technique on the entire input. Each step of the Lloyds-style algorithm is as follows:

- Assign each point to the closest center.

- For each center, solve the 1-median problem on the points that are assigned to it.

We apply the same Idea to our approach as well. We first use our algorithm to create a solution of size $c \cdot k$ for some constant $c$. Then we let the weight of each point in this solution to be the number of points in the subtree in the tree embedding that this points is opened in. Therefore, we achieve a weighted coreset of size $ck$ and then we run greedy improvement on the solution of size $k$ produced by our algorithm.

Here we compare the quality and the running time of this approach with BEL algorithm. We run BEL algorithm on 100 machine, therefore the size of the produced coreset is $100k$. Similarly we let $c = 100$, so the size of our coreset is also $100k$. We run three rounds of greedy improvements for both algorithms. We present the running time of the BEL divided by the running time of our algorithm in Table 4. We also present the quality of the solution in Table 3. We observe that the quality of the solutions are very comparable while our algorithm is noticeably faster.

## E   Description of the Massive Datasets

The synthetic datasets are obtained from the HIGGS dataset. Let $n = 11$ million and $d = 28$ be respectively the number of points and the number of dimensions of the HIGGS dataset. The datasets are generated by copying each point $c$ times. The $i$-th copy of a given point is defined as follows: the first $d$ coordinates are the same as the original point. Then $d + 1$ coordinate is appended and set to $i$ for the $i$-th copy. We create three datasets for $c = 10, 100, 1000$.

## F   Construction of the embedding and missing proofs of Section 3.1

In this section for completeness we present a (standard) random embedding of an input point set with coordinates from $\{0, \ldots, \Delta\}^d$ into a restricted 2-hierarchically separated $T$ (see, for example [32]). We start by describing a deterministic embedding that uses a quadtree structure. We start with a $d$-dimensional axis-aligned cube of side length $\Delta$. This cube corresponds to the root of the tree. We then split the box into $2^d$ subcubes of equal size. Each non-empty subcube corresponds to one node on the next layer of the tree and is connected by an edge of length $\Delta\sqrt{d}/2$, half the maximum

distance of two points inside the cube. Now we apply this procedure recursively: Non-empty cubes[12] of side length $\Delta/2^i$ correspond to nodes in the tree with hop distance $i$ from the root. We subdivide each such cube into subcubes of side length $\Delta/2^{i+1}$ and connect each non-empty subcube to the node corresponding to its containing cube using an edge of length $\sqrt{d}\Delta/2^{i+1}$. The partition stops when the side length of the cube is 1 and so each cube contains a single point. This defines our deterministic embedding into a 2-RHST by mapping each input point to the corresponding leaf. Now we randomize the embedding. This is simply done by adding the same random shift $r$ chosen uniformly at random from $[0, \Delta]^d$ to all input points. Then we employ the above process starting with an initial box of side length $2\Delta$.

Note that every point can compute its position in the embedding independently in fact it only need to have access to its own coordinate and the uniform random shift.

Recall that for two points $p, q \in \{1, \ldots, \Delta\}^d$ we use $\text{DIST}_T(p, q)$ to denote their (shortest-path) distance in the resulting tree $T$. We now show some interesting properties of our embedding.

**Lemma F.1** *Let $p, q \in \{1, \ldots, \Delta\}^d$ be two points. Then*

$$\|p - q\|_2 \leq \text{DIST}_T(p, q) \quad \text{and}$$
$$\mathbf{E}[\text{DIST}_T(p, q)] = O(d \cdot \log \Delta) \cdot \|p - q\|_2.$$

*Proof.* We observe that if $p = (p_1, \ldots, p_d)$ and $q = (q_1, \ldots, q_d)$ are in different subcells of the quadtree of side length $\Delta/2^{i+1}$ then their distance in the tree is at least $\sqrt{d}\Delta/2^i$. Next we observe that there exists a coordinate $i$ such that $|p_i - q_i| \geq \|p - q\|_2/\sqrt{d}$. This implies that for every $i$ with $\Delta/2^{i+1} < \|p - q\|_2/\sqrt{d}$ we have that $p$ and $q$ are in different cells, which implies the first inequality.

Next we prove the second item. Observe that we can view our random shift $s$ as independently shifting each dimension by a random value from $[0, \Delta]$. This implies that the probability that two points are separated in dimension $j$ in the quadtree cells of side length $\Delta/2^i$ is $\min\{1, \frac{|p_j - q_j| \cdot 2^i}{\Delta}\}$. By the union bound, the probability that they are separated in any dimension is at most $\frac{\|p - q\|_1 \cdot 2^i}{\Delta}$. If the points are separated in the quadtree cells of side length $\Delta/2^i$ then there are two edges of length $\sqrt{d}\Delta/2^{i-1}$ on their unique connecting path. We obtain for $h = \log \Delta + 1$ that

$$\mathbf{E}[\text{DIST}_T(p, q)] \leq 2 \sum_{i=0}^{h} \frac{\sqrt{d} \cdot \Delta}{2^i} \cdot \frac{\|p - q\|_1 \cdot 2^i}{\Delta}$$

$$\leq 2 \cdot (h + 1) \cdot \sqrt{d} \cdot \|p - q\|_1.$$

Finally, the result follows since $\|p - q\|_1 \leq \sqrt{d}\|p - q\|_2$. □

**Theorem 3.1** *Let $P \subseteq \{0, \ldots, \Delta\}^d$ be a point set. There exists a procedure that constructs a 2-RHST tree $T$ in time $O(nd \log \Delta)$ such that for the its optimum solution $C^*_{T,k}$ using $k$ centers, we have $\mathbf{E}[\text{COST}(P, C^*_{T,k})] = O(d \cdot \log \Delta) \cdot \text{OPT}(P, k)$. Furthermore all the input points are mapped to leaves of the RHST. In addition, we have that $\mathbf{E}[\max_k \frac{\text{COST}(P, C^*_{T,k})}{\text{OPT}(P,k)}] = O(d \cdot \log \Delta \log(dn\Delta))$.*

*Proof.* Let $C^*$ be the optimal solution on the input data set. By the second item of the previous lemma and linearity of expectation we get $\mathbf{E}[\text{COST}_T(P, C^*)] \leq O(d \log \Delta) \cdot \text{OPT}(P, k)$. By the first item of the previous lemma we then get $\mathbf{E}[\text{COST}(P, C^*_{T,k})] \leq \mathbf{E}[\text{COST}_T(P, C^*_{T,k})] \leq \mathbf{E}[\text{COST}_T(P, C^*)] = O(d \cdot \log \Delta \cdot \text{OPT}(P, k))$.

In order to prove the second statement, observe that the cost of any non-trivial solution is at least 1 and at most $d\Delta n$. Let $k_1, k_2, \ldots, k_m$ be the sequence such that $k_i$ is the largest number of centers such that the optimal $k_i$-median cost is at least $2^i$. We observe that the expected maximum of the $m$ non-negative random variables $\text{COST}(P, C^*_{T,k_i})/\text{OPT}(P, k)$ is at

---

[12]We ignore here degenerate cases that may arise when a point lies on the boundary of the cell, since in the final embedding we will randomly shift the point set and this event will happen with probability 0.

most $m \cdot \max_k \mathbf{E}[\text{Cost}(P, C^*_{T,k_i})/\text{Opt}(P, k_i)]$. By the first item of the theorem this quantity is $O(md \log \Delta)$. Now observe that for any $i$ and any $k$ with $k_i > k > k_{i-1}$ we have $\text{Cost}(P, C_{T,k}) = O(\text{Cost}(P, C_{T,k_{i-1}}))$ since $\text{Cost}(P, C_{T,k})$ is non-increasing in $k$ and we have $2\text{Opt}(P, k) \leq \text{Opt}(P, k_i)$ by the choice of the $k_i$. This implies that $\text{Cost}(P, C^*_{T,k})/\text{Opt}(P, k) = O(\text{Cost}(P, C^*_{T,k_i})/\text{Opt}(P, k_i))$. This implies the second part of the theorem.

Finally we note that the running time is $O(nd \log \Delta)$, since it consist of $\log \Delta + 1$ times of mapping all the points to the nodes of the tree. Also mapping each node is simply dividing all its coordinates by $2^i$ for layer $i$. We also note that this mapping is easily parallelizable because, in order to compute the mapping for a specific node, we only need to know its coordinates and the random shift $r$. □

## G   Missing proof of Section 3.2.1

**Theorem 3.2** *For any set of points $P \subseteq \{0, \dots, \Delta\}^d$ and distance function $\text{Dist}_T$ defined by an 2-RHST $T$, Algorithm 1 returns an optimum solution for the hierarchical $k$-median problem.*

*Proof.* The proof is by induction on the height of the tree $T$. For the base case, when $T$ is of height 0, the statement is clear: in that case the instance consists of a single point so $\text{Cost}_T(P, S) = 0$ if $S \neq \emptyset$.

For the inductive step, suppose that the statement is true for all RHSTs of height less than $h$ and consider an RHST $T$ of height $h$. We use the following notation:

- Let $C$ be the set of children of the root of $T$.
- For a child $y \in C$, let $T_y$ denote the sub-tree rooted at $y$, let $P_y$ denote the subset of leave-nodes in $P$, and let $p_y = |P_y|$.
- Finally, let $D$ be the sum of edge-lengths from the root to a leaf in $T$.

A key observation is that, by the definition of RHSTs, we have $\text{Dist}_T(x, x') = 2D$ for points $x \in P_y$ and $x' \in P_{y'}$ from different sub-trees $y \neq y' \in C$. Therefore, when Algorithm 1 selects $g_y$ centers in the sub-tree $T_y$ for some child $y \in C$, it makes the same selection as if it was ran on the instance consisting of only the points in $P_y$. Now, fix a child $y \in C$ and consider running Algorithm 1 on the subinstance (corresponding to the sub-tree $T_y$) defined by the set of points $P_y$, cost function $\text{Cost}_{T_y}$, and the number of centers equal to $p_y$(Note that in this instance the number of centers is equal to the number of nodes in $P_y$). Let $x_1^{(y)}, \dots, x_{p_y}^{(y)}$ denote the returned centers indexed in the order they were selected by the greedy algorithm. Then, if we let $g_y$ be the number of centers that the greedy Algorithm 1 selects in $P_y$, the solution returned by Algorithm 1 on the whole instance equals $\bigcup_{y \in C: g_y > 0}\{x_1^{(y)}, \dots, x_{g_y}^{(y)}\}$. Moreover, by the induction hypothesis (using that $T_y$ has height $h-1$), we have that $\{x_1^{(y)}, \dots, x_i^{(y)}\}$ is an optimal selection of $i \in \{1, 2, \dots, p_y\}$ centers to the instance on points $P_y$ and cost metric $\text{Cost}_{T_y}$. Now another important observation, that follows from the definition RHSTs, is that the cost of a solution $S$ decomposes:

$$\text{Cost}_T(P, S) = \sum_{y \in C: S \cap P_y \neq \emptyset} \text{Cost}_{T_y}(P_y, S \cap P_y) +$$

$$+ 2D \cdot \sum_{y \in C: S \cap P_y = \emptyset} p_y \, .$$

Thus an optimal solution must optimally select the centers in each subinstance corresponding to a sub-tree. Therefore, since greedy is optimal on each subinstance, any solution that for $y \in C$ selects $o_y$ centers[13] from sub-tree $T_y$ has cost at least $\text{Cost}_T\left(P, \bigcup_{y \in C: o_y > 0}\{x_1^{(y)}, \dots, x_{o_y}^{(y)}\}\right)$. To understand the cost of such a solution, for $y \in C$ and $i \in \{1, \dots, p_y\}$, define the "cost decrease" achieved by adding the center $x_i^{(y)}$ to the set $\{x_1^{(y)}, \dots, x_{i-1}^{(y)}\}$ of centers to be

$$d_i^{(y)} = \text{Cost}_{T_y}(P_y, \{x_1^{(y)}, \dots, x_{i-1}^{(y)}\}) - \text{Cost}_{T_y}(P_y, \{x_1^{(y)}, \dots, x_{i-1}^{(y)}, x_i^{(y)}\}),$$

---

[13]Note that $o_y$ can potentially be different than $g_y$ because $o_y$ is the number of centers in $y$ selected by the optimal solution while $g_y$ is the number of centers selected by the greedy algorithm.

where for notational convenience we let $\text{COST}_{T_y}(P_y, \emptyset) = 2D \cdot p_y$. Then,

$$\text{COST}_T\left(P, \bigcup_{y \in C : o_y > 0} \{x_1^{(y)}, \ldots, x_{o_y}^{(y)}\}\right) =$$

$$= 2D \cdot |P| - \sum_{y \in C} \sum_{i=1}^{o_y} d_i^{(y)}.$$

Now to conclude the inductive step, observe that the greedy selection criteria implies that $d_1^{(y)} \geq d_2^{(y)} \geq \ldots \geq d_{p_y}^{(y)}$ for $y \in C$. Hence, the greedy algorithm select in each subtree a solution of minimum cost. But now using the fact that the cost of solution can be expressed as cost of the solution in the subtrees it follows that Algorithm 1 returns a solution that contains the centers in an order that maximizes at every step the value $d_i^{(y)}$ which, by the above expression, minimize the the cost.

Finally, we note that the partitions induce a hierarchical clustering because every time we open a new center all the points in the sub-tree rooted at the new label node are assigned to it. In fact, suppose that this is false then there is another node in the sub-tree that is already label. But this is impossible by construction because we always label the highest unlabelled ancestor in the tree and so an already labelled node cannot be at a lower level of a newly labelled node. □

# H    Missing proof of Section 3.2

We start by showing the. following lemma:

**Lemma H.1** *Algorithm 2 and Algorithm 1 output the same solution.*

*Proof.* Suppose by contradiction that the two algorithms return different ordering of the centers. In particular, let $\{v_1^1, v_2^1, \ldots\}$ the ordering in which points are selected by Algorithm 1 and $\{v_1^2, v_2^2, \ldots\}$ the ordering obtained after sorting by Algorithm 2 and let $i$ be the first index for which $v_i^1 \neq v_i^2$. Recall that during the execution of the two algorithms we sort by breaking ties consistently(for example by looking at points ids). Our proof strategy is to show that if $v_i^1$ is selected by Algorithm 1 over $v_i^2$, this contradicts the fact that $v_i^2$ is the $i$-th element in the ordering Algorithm 2.

Now, let $\ell$ be the largest number such that $A_\ell(v_i^1)$ does not contain any node in $\{v_1^2, v_2^2, \ldots, v_{i-1}^2\}$ and let $\ell'$ be the largest number such that $A_{\ell'}(v_i^2)$ does not contain any node in $\{v_1^2, v_2^2, \ldots, v_{i-1}^2\}$. We note that there are two nodes $v_x \in A_\ell(v_i^1)$ and $v_y \in A_{\ell'}(v_i^2)$ for which $\text{BENF}(v_x, \ell')$ and $\text{BENF}(v_y, \ell')$ have not been discarded by Algorithm 2. This is true because in the first filtering Algorithm 2 does not discard the maximum benefit elements in the subtrees rooted $a_\ell(v_i^1)$ and $a_{\ell'}(v_i^2)$. And in addition, at any level larger than $\ell(\ell')$ no element in $A_\ell(v_i^1)$ ($A_{\ell'}(v_i^2)$) is the element of maximum benefit by definition of $\ell$ and by the fact that Algorithm 2 selected $\{v_1^2, v_2^2, \ldots, v_{i-1}^2\}$. So the maximum benefit elements in the subtrees rooted $a_\ell(v_i^1)$ and $a_{\ell'}(v_i^2)$ are not discarded even in this second phase.

Now, if $v_y \neq v_i^2$ Algorithm 2 would. have select $v_y$ over $v_i^2$ because after filtering we know by the above observations that $v_y$ is the element with maximum benefit in $A_{\ell'}(v_i^2)$. For the same reason we also have that $\text{BENF}(v_x, \ell) \geq \text{BENF}(v_i^1, \ell)$. In addition given that Algorithm 2 selects $v_i^2$ as $i$-th center we have $\text{BENF}(v_i^2, \ell') \geq \text{BENF}(v_x, \ell) \geq \text{BENF}(v_i^1, \ell)$. But now, by the definition of $\text{BENF}$ and by observation 3.3 and lemma 3.4 this imply that the cost of $\{v_1^1, v_2^1, \ldots, v_{i-1}^1, v_i^2\}$ is smaller than the cost $\{v_1^1, v_2^1, \ldots, v_{i-1}^1, v_i^1\}$ or they are the same but the ties are broken in favor of $v_i^2$. So Algorithm 1 will select $v_i^2$ over $v_i^1$ leading to a contradiction. So the two orderings and the two solutions are identical.

For the partition we note that the also in the new algorithm the points are assigned to the closest labelled ancestor and so the partitions are equivalent. □

We are now ready to show that Algorithm 2 has the desired properties.

**Theorem 3.5** *Algorithm 2 finds an optimum solution for the hierarchical $k$-median problem on RHSTs in time $O(n \log^2(\Delta + n))$.*

*Proof.* The fact that Algorithm 2 finds the optimum solution follows from combining Lemma H.1 and Theorem 3.2. Furthermore, all the BENF (Line 1) can be compute in time $O(n \log^2 \Delta)$, since we have to compute $O(n \log \Delta)$ many and each is a sum of $O(\log \Delta)$ values. Now consider some level $\ell$ in the tree embedding. The total overall number of pairs $(\cdot, \ell)$ in each layer is exactly $n$. Therefore computing the maximum (Line 4), deleting the rest (Line 6) and construct the sets $C_{(x,\ell)}$ takes $O(n)$ time for each layer and $O(n \log \Delta)$ time in total. The second filtering step (Line 10) can also be done in time $O(n \log \Delta)$ since it finds the maximum elements in $n$ lists (one for each element) and each list is of length $O(\log(\Delta))$. Finally, we have that after filtering at most one pair $(x, \cdot)$ for each point $x$ remains. So we can sort the remaining pairs (at most $n$ many) in time $O(n \log n)$, which concludes the lemma. $\square$

## I Results in the General Space

In this section we address the assumption over the space of the input points, $P \subseteq \{0, \ldots, \Delta\}^d$.

First observe that given a general instance $P \subseteq \mathbb{R}^d$ and a rough estimate of the optimum solution (poly$(n)$), one can achieve an instance $P \subseteq \{0, \ldots, \Delta\}^d$ for some $\Delta \in \text{poly}(n)$ by losing a factor $1 + 1/\text{poly}(n)$ in the approximation guarantee in linear time. Unfortunately such estimate of the optimum solution might not be achievable in some cases. Therefore we explain an alternative idea.

The only part in our approach that is using the assumption of $P \subseteq \{0, \ldots, \Delta\}^d$ is the tree embedding step, i.e., 2-RHST. We present how to construct such embedding for $P \subseteq \mathbb{R}^d$. To this end, we first compute an upper bound MAXDIST on the maximum distance between two points within a factor of 2. This can be done by selecting any point and by computing the maximum distance between such point and any other point in the input. Then multiply this distance by 2. The root of the tree represents an axis-aligned cube of side length 2MAXDIST centered at $x$, we apply the same construction as before to construct each height of the 2-RHST. We continue until every cube contains at most a single point. This results in a tree where all leaves are at the same height, the height is at most $H = O(\log(d\Delta'))$ where $\Delta'$ is the ratio between the maximum and minimum distance of two points. Moreover to achieve that the length of the edges at height $\ell$ is $2^\ell$, we can divide the value of all the edges in the tree by the length of the edges at height zero. Recall that all the edges at the same height have the same value.

This ideas enables to get the results presented in the paper with the same bound but $\Delta$ is replaced by $d\Delta'$. Notice that since in all the bounds $\Delta$ only appears in log function, this change does not affect the performance significantly.

## J Reducing Dimension in Distributed Setting

The dimensionality reduction is obtained by basically multiplying an $n$-by-$d$ matrix (representing the input points) with a random $d$-by-$O(\log n)$ matrix. This can be done using memory $s \in \Omega(d \log n)$ in $O(\log_s(nd \log n))$ MPC rounds with $O((nd \log n)/s)$ machines and a total running time of $O(nd \log n)$ similar to algorithms in [28].

## K Parallel Speed-up

In this section, we describe our empirical results on the speed-up obtained by increasing the number of machines. We consider artificial datasets of sizes 100 millions, 1 billion and 10 billions points. We give the relative running time as the number of machine increases in Table 5.

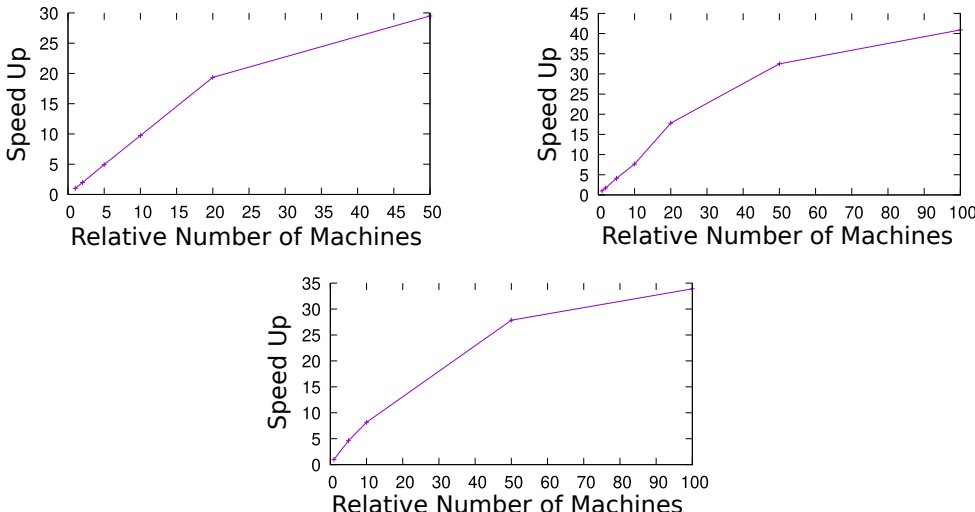

Figure 5: Speed-up for the massive datasets of size almost 100 millions, 1 billion, and 10 billion.