# OpenReview forum: "Parallel and Efficient Hierarchical k-Median Clustering"
_NeurIPS.cc/2021/Conference — NeurIPS 2021 Poster_

### Official Review · Reviewer_2DG5 · 2021-07-14

**Rating:** 7
**Confidence:** 3

**Summary:**

The authors of this paper consider the Hierarchical k-median clustering problem in the Euclidean setting, and they give approximation algorithms with provable guarantees for it in the Massive Parallel Computation (MPC) model. Specifically, they given the first results with bounded approximation ratios for the problem in the MPC model.

Although I'm somewhat familiar with hierarchical clustering, this is the first time I came across the MPC model. However, it looks like a very reasonable setting for parallel computation. According to it, each machine has memory $s$. Then we proceed with a number of rounds, in which each machine processes data locally, and at the end of the round the machines can exchange messages of size at most $s$. Obviously, efficiency in this setting is measured with respect to how large $s$ should be and how many rounds the algorithm might take. The authors study an interesting and realistic I would say regime, where the memory available to each machine is relatively small.

The authors solve their problem in two steps. First they provide a sequential algorithm. The latter begins by embedding the metric space in a a restricted 2-hierarchically separated tree (2-RHST). Then the authors show how you can find an optimal solution for an 2-RHST instance, and hence in the sequential step they end up only paying the distortion of the embedding in the approximation ratio. Finally, for the second step of their framework they show how their sequential algorithm can be readily parallelized in MPC model.

Finally, experiments are presented, which clearly demonstrate that the given algorithm is much faster than many possible baselines (both parallel and sequential heuristics). In addition, in terms of the objective function value, it is shown that the output of the algorithm is actually quite comparable to even sequential approaches like k-median++. Another experimental advantage of the proposed algorithms, seems to be its simplicity that leads to it being easily implemented.

**Limitations And Societal Impact:**

Yes

**Main Review:**

CONFIDENCE - ORIGINALITY:

Although I'm familiar with metric clustering, I have to admit that I do not have much experience in hierarchical clustering. In addition, I'm completely unfamiliar with the literature on parallel algorithms. That being said, I do not believe I can adequately judge the originality and the significance of this work. Nonetheless, it feels like providing the first parallel algorithms for hierarchical k-median is a solid contribution and that's why I believe that this paper should appear in NeurIPS.

CLARITY:

For the most part the paper is nicely written and clear. I would have preferred a more detailed discussion on related work on sequential hierarchical k-median. Specifically, an expansion of the paragraph in lines 41-47.

QUESTION/SUGGESTION TO AUTHORS:

My point here relates to my previous comment regarding discussing related work more. Can you elaborate more on the properties of the LLLM algorithm you use as a baseline in Section 5.1? This seems like the most comparable competitor and it feels like you should have explained it in more detail. For example, is it a heuristic (I suppose it is)? What runtime guarantees it has?



**Time Spent Reviewing:**

5 hours reading the paper + 1 hour thinking about the review + 1 hour writing the review

---

> ### Author Response · Authors · 2021-08-06
> **Rebuttal to Reviewer 2DG5**
>
> We thank you very much for your time, the review and the helpful comments.
>
> A short description and details of all the baseline algorithms are explained in the Appendix B. The LLLM algorithm is distributed and hierarchical, it starts with all the points in one cluster and it divides each cluster into two until the size of all the clusters are one. It guarantees that the number of parallel rounds is O(log n∆) but it does not provide a theoretical guarantee on the quality of the solution (approximation guarantee).

---

### Official Review · Reviewer_c7EP · 2021-07-16

**Rating:** 5
**Confidence:** 4

**Summary:**

This manuscript studied introduce a new parallel algorithm for the Euclidean hierarchical k-median problem that claimed to be the first parallel algorithm for the hierarchical k-median problem with theoretical guarantees. However, I slightly recommend for acceptance despite the imperative importance of targeted problem.

**Limitations And Societal Impact:**

Cons:

1.	The proposed model strongly relies on the existence of a good tree embedding, which weaken the applications of the model and uncertain results of specific tasks.
2.	There's no theoretical guarantee that for all k, the cost of the solution is at most a factor bigger that the corresponding optimal solution.
3.	It seems not to be the first distributed approximation algorithm for Euclidean Hierarchical k-Median problem in the MPC model. In my research, it is possible to compute an approximate solution for the Euclidean k-means problem using parallel memory.
4.	There's no theoretical explanation to speed up the algorithm for datasets of large dimension, by reducing the dimension from d by losing a small constant factor in the approximation ratio.
5.	The experiment needs to be further improved and strengthened. All data sets used do not reflect the diversity of data, and obviously cannot reflect all adaptation scenarios in the real world. At the same time, the advantages of the experimental results on some data sets are not prominent.



**Main Review:**

Pros:

1. Clearly written and well-organized manuscript, lucid background and problem statement.
2. The motivation is creative, presenting the first distributed approximation algorithm for Euclidean Hierarchical k-Median problem in the MPC model, when k-median received a lot of attention and the problems have been studied extensively.



**Time Spent Reviewing:**

two

---

> ### Author Response · Authors · 2021-08-06
> **Rebuttal to Reviewer c7EP**
>
> We thank you very much for your time and the review.
>
> 1- The proposed model strongly relies on the existence of a good tree embedding, which weakens the applications of the model and uncertain results of specific tasks.
>
> Notice that such a tree embedding can be computed efficiently for any input embedding, the bounds stated in the paper are for arbitrary inputs.
>
> 2- There's no theoretical guarantee that for all k, the cost of the solution is at most a factor bigger that the corresponding optimal solution.
>
> There might be some confusion here. The theoretical guarantee is for all k. This is formally explained in the introduction lines 123-124. It is also mentioned before the main result Corollary 4.2. We will make it more explicit in the paper.
>
> 3-It seems not to be the first distributed approximation algorithm for Euclidean Hierarchical k-Median problem in the MPC model. In my research, it is possible to compute an approximate solution for the Euclidean k-means problem using parallel memory.
> This paper does present the first approximation algorithm for the Euclidean Hierarchical k-Median problem in the MPC model. We acknowledge that in the literature, there are results showing:
> How to recursively compute a constant factor approximation to k-median or k-means (a.k.a. Bisecting k-means) in parallel environments. However, this does not guarantee any approximation for the clustering obtained for a given k (since it depends on the previous splits done by the algorithm), and so this is incomparable to our result.
> How to approximate the k-means or k-median problem in MPC when the memory per machine is at least k. If the memory per machine is s, it is clear that these algorithms cannot help to compute an approximate solution for k-means or k-median for any k = Omega(s). Thus, if the memory per machine is sublinear in the input size, these algorithms cannot be used to solve the Hierarchical k-median or k-means problems since it requires to solve k-median or k-means for all values of k, up to k equals the input size.
> If the reviewer has a reference that we are missing and that explicitly provides an approximation algorithm for the hierarchical k-median or k-means in the MPC model when the memory per machine is sublinear in the input size, we would be happy to investigate.
>
> 4- There's no theoretical explanation to speed up the algorithm for datasets of large dimension, by reducing the dimension from d by losing a small constant factor in the approximation ratio.
>
> Reducing dimension is discussed in Appendix J. The goal of reducing the dimension is to achieve better approximation guarantee and is not to speed up the algorithm. It is explained in lines 320-324 and Corollary 4.2 formalize it.
>
> 5- The experiment needs to be further improved and strengthened. All data sets used do not reflect the diversity of data, and obviously cannot reflect all adaptation scenarios in the real world. At the same time, the advantages of the experimental results on some data sets are not prominent.
>
> We have used the famous datasets that are commonly used in this area. Our experiments use 8 datasets (5 in the main body and 3 extremely large ones in the appendix). We do not think that adding extra datasets or experiments can improve the message of this paper. For all the datasets our algorithm is significantly faster than the baselines which is the main goal of our paper. Notice that for the large datasets (100M  to 10B points) it is not even possible to run the baselines. Compared to LLLM  our algorithm is faster by at least a factor 6.7 − 26.9, (see Table 1).  This is also the case for the algorithms that do not perform in the hierarchical setting. For small values of k, e.g: k = 10, we are faster by a factor 3.2 − 60.2, 50 − 182, and 65 − 1380 compared to BEL, EIM, and CSZ, respectively. Moreover, for k = 1,000 we are 12.08 − 71.2, 2082 − 3091, 46 − 941 times faster compared to BEL, EIM, and CSZ, respectively. Also for k = 10,000 we are 41.17 − 126 and 171 − 708 times faster than BEL and CSZ, respectively.

---

### Official Review · Reviewer_kqYe · 2021-07-16

**Rating:** 7
**Confidence:** 4

**Summary:**

This paper proposes a parallel algorithm for hierarchical k-means in the MPC model. The algorithm makes use of a randomized tree embeddings of a dataset. The authors prove the approximation ratio for their algorithm (for every value of k) as well as the space and round complexity. In experiments on 4 datasets, the proposed algorithm achieves either competitive/superior performance to a handful of baselines (in terms of clustering cost), and is demonstrated to be scalable and efficient.

**Ethical Concerns:**

I do not see any ethical issues with this work.

**Limitations And Societal Impact:**

As mentioned in the introduction, one limitation is the proposed algorithm relies on a high-quality and efficiently computable tree embedding of a dataset.  Also, as mentioned, their theoretical results only hold in expectation, but this is understandable given their randomized embedding. Also, their bounds are a function of the largest and smallest distance between any two points in the dataset.

Due to space constraints much of the proofs are significantly shortened or completely skipped. It would have been nice to include more of these proofs for a better understanding (for example, for Theorem 3.2).


**Main Review:**

*Originality*:
This work builds on existing research in parallel/distributed clustering. The proposed algorithm is novel, but is built from well-known components: RHST embedding techniques and achieving certain operations in the MPC model.  The key novelty in the algorithm lies in the observations that many of the required computations only require knowledge of the RHST, and can therefore be performed in parallel. The proof techniques are relatively straightforward, but the observations which lead to the parallelization are interesting.

*Quality*:
While there are a lot of details skipped in the proofs, the algorithm and arguments made in this paper seem correct to me. While reading, I had a question about fundamentals (appears in the clarity paragraph), but I have assumed this is my own misunderstanding rather than a flaw.

The experiments are sufficient but could be improved. The proposed algorithm is compared to a handful of baselines in terms of cost and speed. The datasets chosen are reasonable, but better datasets could be used. In particular, none of the datasets have particularly high dimension and only 2 have more than 1M points. Since the theoretical results depend on \delta, it would be informative to compute/report \delta for each of the datasets.

In the experiments, is the time for computing the RHST included in the timing? Please be clear about this; it is an important detail.

*Clarity*:
This paper was a pleasure to read. It is very clear: the writing/grammar is easy to digest, the flow of ideas is sensible, the authors provide examples and high-level descriptions of their algorithms to make their theoretical arguments easy to follow. I especially like the progression from sequential to parallel to MPC, which I thought made the algorithms easily understandable.

One thing that was slightly difficult was keeping track of all the tree notation in 3.2.2.  Perhaps a visual aid could help.

In Section 4: loose → lose

Algorithm 2: after line 10, we have for every subtree, a single point x that maximizes benefit. Then, in line 11, it seems like we filter these points so that each point can only be associated with a single subtree (the highest one where it maximizes benefit). But doesn’t this mean that if a point maximizes benefit for more than 1 subtree, then after filtering, there will be some subtrees with no associated point/center? Please clarify. In writing my review I assume this is my own misunderstanding and not a flaw in the algorithm.

In your proposed algorithm, is the RHST computed in a distributed fashion or prior to the MPC setting?

*Significance*:
Empirically, the proposed algorithm has potential to be adopted in practice. The empirical results demonstrate that the algorithm achieves either superior or competitive performance with a number of baselines (and has relatively small variance). Moreover, it is scalable and efficient, which are both very important given the increasing size of datasets and computing infrastructure.

Algorithmically, this work is also significant. While the idea of embedding a dataset in an RHST is not new, the parallelization of the clustering algorithm after this transformation seems novel. As the authors mention in their conclusion, it may be possible to apply similar techniques to develop additional parallel clustering algorithms for different cost functions--which would be useful.

While many of the proofs are skipped, I do not think that any significantly new proof techniques were developed in this work.

EDIT AFTER AUTHOR RESPONSE: thank you for the informative response, especially the pointers to the appendix and the clarifications about not explicitly computing the RHST and the experiments/datasets/baselines. Also, thank you for fixing the typos in the algorithm.

**Time Spent Reviewing:**

2.5

---

> ### Author Response · Authors · 2021-08-06
> **Rebuttal to Reviewer kqYe**
>
> We thank you very much for the time spent, the review and the helpful comments. The comments that are not addressed below will be implemented in the camera ready version.
>
> Regarding “The experiments are sufficient but could be improved. The proposed algorithm is compared to a handful of baselines in terms of cost and speed. The datasets chosen are reasonable, but better datasets could be used. In particular, none of the datasets have particularly high dimensions and only 2 have more than 1M points. Since the theoretical results depend on \delta, it would be informative to compute/report \delta for each of the datasets.”
>
> We would like to point out that the baselines are slow compared to our algorithm and it is not possible to run them on large datasets (we tried and they failed for 110M points). We have further experiments that are reported in the appendix. In Appendix K, we run experiments on our algorithm for datasets of size 100 million, 1 billion and 10 billions as well.  In Appendix C, we report further sequential results. In Appendix D we present results for the case that our algorithm is used to construct coresets.
>
> Regarding “In the experiments, is the time for computing the RHST included in the timing? Please be clear about this; it is an important detail.”
> Yes, of course. Notice that we do not explicitly construct it (details are explained in the proof of Theorem 4.1). We will clarify this in the camera ready version.
>
> Regarding “Algorithm 2: after line 10, we have for every subtree, a single point x that maximizes benefit. Then, in line 11, it seems like we filter these points so that each point can only be associated with a single subtree (the highest one where it maximizes benefit). But doesn’t this mean that if a point maximizes benefit for more than 1 subtree, then after filtering, there will be some subtrees with no associated point/center? Please clarify. In writing my review I assume this is my own misunderstanding and not a flaw in the algorithm.”
> There is a small typo in the algorithm, line 11 should be executed before line 2. The algorithm explanation in the paper, proofs and statements are correct. We apologize for the typo.
>
> Regarding “In your proposed algorithm, is the RHST computed in a distributed fashion or prior to the MPC setting?”
> There is no need to compute the RHST explicitly but it can be implemented efficiently in the  distributed setting as well. On page 8, in line 305-319 the details of the algorithm is explained.
>
> Regarding “As mentioned in the introduction, one limitation is the proposed algorithm relies on a high-quality and efficiently computable tree embedding of a dataset.“
> We do not think of this as a limitation, it is simply a step in our algorithm. All the guarantees and running time reported take this step into account and the tree embedding part of the algorithm is not used as a black box.

---

### Official Review · Reviewer_zNbV · 2021-07-17

**Rating:** 4
**Confidence:** 3

**Summary:**

The paper proposes algorithms for hierarchical k-median clustering. Besides my concern with its technical quality, I had a hard time following some of the statements/proofs in the paper.


**Ethics Review Area:**

["I don’t know"]

**Limitations And Societal Impact:**

The paper does have a section discussing limitations: that is relies on a good tree embedding that can be computed fast.

**Main Review:**

I am not too familiar with the recent advances of hierarchical clustering. I feel the authors should include some general introduction on how tree embedding is used here: What would be the main motivation for using the quad tree embedding technique needed here?  The paper mentions that the embedding from P will only incur a small distortion. Then what would be the benefit of using such embedding technique? How is it different from directly using Euclidean space metric? Is it because the assumption that the dataset P are sitting on integer grid {0, ..., \Delta}^d? If so, what's the motivation behind this assumption?

For Theorem 3.2, the proof sketch argues that the cost can be decomposed as the cost of subtrees. I don't see how this is true for the k-median cost: for example, after finding the optimal (or near-optimal) solution for k=1; when k=2, shouldn't algorithm consider to move the first center as well? Can the authors provide more details on how the greedy algorithm work here?





**Time Spent Reviewing:**

4hrs

---

> ### Author Response · Authors · 2021-08-06
> **Rebuttal to Reviewer zNbV**
>
> We thank you very much for the time spent, and the review. We start by addressing the main question
>
> Regarding “What would be the main motivation for using the quad tree embedding technique needed here?” The paper mentions that the embedding from P will only incur a small distortion while providing a very structured representation of the input distances (in the form of a tree), that enable efficient, massively parallel algorithms.
>
> Then “what would be the benefit of using such embedding techniques? How is it different from directly using the Euclidean space metric? Is it because of the assumption that the dataset P is sitting on an integer grid {0, ..., \Delta}^d? If so, what's the motivation behind this assumption?”
>
> No, it is not related to this assumption. Notice that in the appendix we discuss options to remove this assumption. The assumption is just made in the main body for sake of simplicity. We use the tree embedding because then on the tree we can solve the problem optimally and efficiently. In fact, the greedy algorithm discussed in section 3 only returns an optimum solution for the tree metric, and may provably return solutions of poor quality if using the Euclidean distances directly. Moreover the distributed algorithm explicitly uses the tree embedding and cannot be executed for general metric.
>
> Regarding “For Theorem 3.2, the proof sketch argues that the cost can be decomposed as the cost of subtrees. I don't see how this is true for the k-median cost: for example, after finding the optimal (or near-optimal) solution for k=1; when k=2, shouldn't the algorithm consider moving the first center as well? Can the authors provide more details on how the greedy algorithm works here?”
> No, the algorithm does not need to consider moving the centers after they are chosen. The main reason is that the distance between the points is defined to be the shortest path on the tree embedding. A complete detailed proof is provided in Appendix G.

---

> ### Comment · Reviewer_zNbV · 2021-09-14
> **Unable to judge the quality of the paper**
>
> After reading authors' response and other reviewer comments, I'm still not able to judge the quality of the paper. I have mainly two concerns: 1. In the main paper, the setup of the problem is not clearly defined 2. In the main paper, the proofs provided do not seem very rigorous to me.
> In addition, I do not have access to the supplementary/Appendix for this paper, so I cannot carefully read the full theoretical arguments. However, given that most other reviewers give a high score (or at least higher than mine) to the paper, I'm okay with the paper being accepted.

---

> > ### Author Response · Authors · 2021-09-14
> > **Points about the concerns**
> >
> > Thanks for the comment, we are sorry for the difficulty that you experienced. The model is formally described in the main body (please refer to line 33 for definition of hierarchical k-median and line 66 for the definition of the distributed setting). As mentioned in the paper, some of the proofs are omitted in the main body and only a sketch is provided. The complete proofs can be found in The appendix. In order to download it, after choosing this paper in the reviewer console, there is a zip icon in the "Supplementary Material:" field. By clicking on it you get a zip folder that contains the entire paper and a zip code folder.

---

### Official Review · Reviewer_KiLL · 2021-08-01

**Rating:** 7
**Confidence:** 4

**Summary:**

This paper studies the hierarchical k-medians clustering problem and develops a new parallel algorithm. The algorithm is based on building a tree embedding of the dataset parallelly first and then the partitions of the dataset are computing using this tree. A rigorous theoretical analysis is provided for the communication complexity, memory, and the approximation guarantees of the algorithm for the k-medians problem for all k. The provided experiments complement the theoretical results.

**Ethical Concerns:**

None.

**Limitations And Societal Impact:**

Some potential limitations are discussed.
This paper focuses solely on the k-medians problem. Why this algorithm is specific to the k-medians objective and what is stopping one from using a similar technique for other k-clustering problems can be an important discussion to have.

**Main Review:**

# Originality

Although tree embedding-based methods are popularly used in distributed algorithms, this paper develops the first theoretical guarantees for the parallel hierarchical k-medians clustering problem using tree-based partitions. I believe a substantial amount of original work is needed in the analysis and developing correct tools to build the algorithm.

# Quality

The writing and the presentation of the paper are good in general.  Motivations are discussed adequately and this result is placed well among the other related work. Detailed experiments provide a good understanding of how the algorithm works and its performance.

# Clarity

The clarity of the paper for the most part is good. However, I have some clarifying questions.
- In the abstract, the authors claim that "for all k simultaneously the cost of the solution is at most $O(min (d, log n) \log \Delta \log (\Delta dn))$ factor bigger than the corresponding optimal solution", however, if I have not mistaken, the provided theorem(theorem 4.1) only gives a result for every fixed k in expectation. Is the general result true with high probability? Can the authors clarify more on this?

# Significance

I believe the result presented in this paper carries a high level of significance given that it provides the first theoretical guarantees for parallel hierarchical k-medians and the approximation guarantees for the objective are not too far from the standard k-median++(which is a serial algorithm) guarantees. This algorithm also avoids the memory restrictions per node (here, only a small amount of memory is required) which was considered a key bottleneck in prior coreset based distributed algorithms for k-medians(needs to store at least k points per machine). In addition, the algorithm also matches the known round lower-bounds.

Overall, I believe this paper adds very nice contributions.


**Time Spent Reviewing:**

3

---

> ### Author Response · Authors · 2021-08-06
> **Rebuttal to Reviewer KiLL**
>
> We thank you very much for the review, the time spent and the helpful comments.
>
> Regarding your comment about the approximation guarantee. Our algorithm indeed satisfies the following property, with probability at least 1/2 the following holds: for all values of k simultaneously, the algorithm outputs a solution that is within a O(min {d, log n} log ∆ log(dn∆)) factor of the optimum solution for the k-median. We will make this explicit in theorem 4.1 as well that the result is for all the $k$ values, i.e.: the maximum approximation factor for all $k$ values in range $1$ to n is O(min {d, log n} log ∆ log(dn∆)) in expectation (exactly the same as theorem 3.1). We will also fix the typo in the abstract

---

> > ### Comment · Reviewer_KiLL · 2021-09-01
> > **Thank you**
> >
> > I thank the authors for the clarification. I have read the other reviews and the author's responses. I maintain my evaluation that this paper adds significant contributions.
> > As mentioned please mention the details explicitly in the theorem statements.

---

### Decision · Program_Chairs · 2021-09-27

**Decision:**

Accept (Poster)

**Comment:**

The authors present an algorithm for the distributed hierarchical k-median problem that requires only logarithmic memory on the machines, logarithmic rounds of communication, and has an O(d*log(n)) approximation factor for each k simultaneously. The approach taken combines tree embeddings with a careful parallelization of the clustering operations. The algorithm is of practical interest to the community, and the approach may lead to further innovation and efficient approaches to other clustering problems.